# Optimisation of Supercritical CO$_2$ Extraction from Black (*Ribes nigrum*) and Red (*Ribes rubrum*) Currant Pomace

Filip Herzyk and Małgorzata Korzeniowska *

Department of Functional Food Products Development, Faculty of Biotechnology and Food Sciences, University of Environmental and Life Sciences, 51-630 Wrocław, Poland; filip.herzyk@upwr.edu.pl
* Correspondence: malgorzata.korzeniowska@upwr.edu.pl

**Featured Application**

The results of this study highlight the potential of using supercritical CO$_2$ extraction to valorise berry pomace as a source of functional ingredients. The obtained lipid-rich and phenolic-compound-rich fractions may serve as valuable additives in the development of nutraceuticals, functional foods, and natural cosmetics. Further research should focus on the bioavailability and stability of these extracts in complex formulations and evaluate their efficacy in vivo.

**Abstract**

Fruit pomace, generated as a by-product of juice processing, is a valuable source of bioactive compounds but requires sustainable extraction approaches to enable its valorisation. Supercritical CO$_2$ extraction (SFE-CO$_2$) represents a promising green technology due to its efficiency, solvent-free character, and tuneable selectivity. In this study, the response surface methodology (RSM) was applied to evaluate the effects of pressure, temperature, and time on the recovery of fat, protein, and total phenolic compounds (TPCs) from blackcurrant (*Ribes nigrum*) and redcurrant (*Ribes rubrum*) pomace subjected to conventional- and freeze-drying. The highest protein content (14.5%) was obtained in freeze-dried blackcurrant at 400 bar, 60 min, and 30 °C, while the maximum TPCs (24.60 mg GAE/g d.w.) was reached at 500 bar, 60 min, and 40 °C. The redcurrant samples consistently showed lower extractable values across all the responses. Pressure and time were identified as the most influential process variables, enhancing the solvent density and mass transfer during extraction. These results demonstrate that both the drying pre-treatment and raw material type significantly affect the SFE efficiency and confirm the potential of optimised SFE-CO$_2$ as a viable strategy for converting fruit pomace into functional ingredients for food, nutraceutical, and cosmetic applications.

**Keywords:** supercritical extraction; pomace; blackcurrant; redcurrant; lyophilisation; by-products

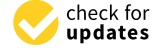

## 1. Introduction

Supercritical fluid extraction (SFE), particularly using carbon dioxide (SFE-CO$_2$), is recognised as a promising green technology for recovering bioactive compounds from plant-based biomass [1,2]. Compared with conventional solvent extraction methods, SFE offers several advantages, such as lower organic solvent consumption, shorter processing time, reduced energy demand, and improved selectivity through precise control of pressure and temperature [3–5]. Furthermore, the ability to easily remove the solvent by

decompression minimises the risk of residual contaminants, making SFE extracts suitable for food, pharmaceutical, and cosmetic applications [4,6].

SFE plays an important role in circular economy strategies, enabling the valorisation of agro-industrial by-products through environmentally friendly processes. For instance, studies on berry seeds and skins—such as raspberry, blueberry, pomegranate, blackberry, and blackcurrant—demonstrated that SFE produces purer lipid fractions with higher polyunsaturated fatty acid (PUFA) levels than traditional hexane-based extraction [2,6]. Fruit pomace, generated during juice and fruit processing, accounts for 10–60% of the raw material weight, with its global production estimated at approximately 0.5 billion tons annually [7,8]. Common disposal practices, including landfilling or incineration, contribute to environmental pollution and the loss of valuable bioactive compounds [2,9,10]. Recent studies have shown that pomace retains significant amounts of proteins, lipids, polysaccharides, dietary fibre, vitamins, phenolic compounds, pigments, and aromatic substances [11]. Berry seeds, in particular, are rich in oils containing essential fatty acids, vitamin E, and antioxidant compounds [12,13].

Due to their high moisture content, pomaces are highly perishable and prone to microbial spoilage, making drying a critical step for stabilisation. Drying reduces water activity, extends shelf life, and facilitates handling and further processing. Conventional hot-air-drying is widely used in the food industry due to its simplicity and relatively low cost; however, prolonged exposure to heat and oxygen can cause oxidative reactions and degrade thermolabile bioactive substances [14,15]. Freeze-drying (lyophilisation) is considered the most effective dehydration method for preserving structural integrity and sensitive compounds, owing to the low temperatures, vacuum conditions, and minimal enzymatic activity [16]. Despite its advantages, freeze-drying is associated with high energy consumption—up to 4–10 times greater than conventional drying—and long processing times, which limit its industrial-scale application [16]. Nevertheless, it produces high-quality materials with retained bioactivity, which are commonly used as functional food ingredients, nutraceuticals, and additives in instant products, bakery goods, and beverages [17].

Among the berry pomaces, blackcurrant (*Ribes nigrum*) residues are particularly valuable due to their high levels of anthocyanins [7], flavonoids, and vitamin C [18,19], which provide strong antioxidant, anti-inflammatory, and cardioprotective effects [20]. The seeds contained in blackcurrant pomace are a source of oil rich in PUFA, making them suitable for food and cosmetic applications [10]. Recent studies have demonstrated that supercritical $CO_2$ extraction of blackcurrant seeds under pressures of 230–300 bar and 40 °C ensures efficient oil recovery with high nutritional quality, offering a viable alternative to cold pressing [10]. These trends highlight the growing demand for sustainable processing technologies that combine waste valorisation with the production of high-value functional ingredients [21].

The objective of this study was to optimise the supercritical $CO_2$ extraction (SFE-$CO_2$) conditions for the recovery of bioactive compounds from dehydrated blackcurrant (*Ribes nigrum*) and redcurrant (*Ribes rubrum*) pomace. Response surface methodology (RSM) was applied to evaluate the effects of pressure, temperature, and time on the extraction yield of fat, protein, and total phenolic compounds.

## 2. Materials and Methods

### 2.1. Raw Materials

Fresh blackcurrant pomace (*Ribes nigrum*) was obtained from two sources: Wiatrowy Sad Juice Pressing Facility (Wiatrowy Sad Grażyna Wiatr, Dmosin, Poland) and Maspex

Group (Wadowice, Poland). Redcurrant pomace (*Ribes rubrum*) was acquired from Wiatrowy Sad Juice Pressing Facility (Wiatrowy Sad Grażyna Wiatr, Dmosin, Poland).

## 2.2. Equipment

All the technological processes and laboratory analyses were conducted using equipment available at laboratories in Wroclaw, particularly those of the Wroclaw University of Environmental and Life Sciences. Supercritical fluid extraction was carried out using a three-basket extractor with a total volume of 10 L (Natex Prozesstechnologie GesmbH, Ternitz, Austria), operating in a closed $CO_2$ loop. Lyophilisation was performed using a Martin Christ Delta 1-24 LSC freeze dryer (Martin Christ Gefriertrocknungsanlagen GmbH, Osterode am Harz Germany) in combination with a low-temperature freezer (New Brunswick Premium U410, New Brunswick Scientific Co., Inc., Trenton, NJ, USA). For comparative purposes, conventional low-temperature-drying was conducted in a chamber dryer (EL-TECH, Wieluń, Poland).

## 2.3. Research Methodology

The raw material was divided into two primary fractions based on the drying method: fraction (1) was designated for conventional-drying and fraction (2) for freeze-drying. Following dehydration, each dried material was subjected to supercritical $CO_2$ extraction, resulting in two secondary fractions: fraction (3), derived from conventionally dried material (1), and fraction (4), derived from freeze-dried material (2). The extraction also yielded a lipid-like extract fraction, which was collected for potential future analyses but is not considered in the present study. The overall experimental workflow, including the drying and subsequent supercritical $CO_2$ extraction steps, is illustrated in Figure 1.

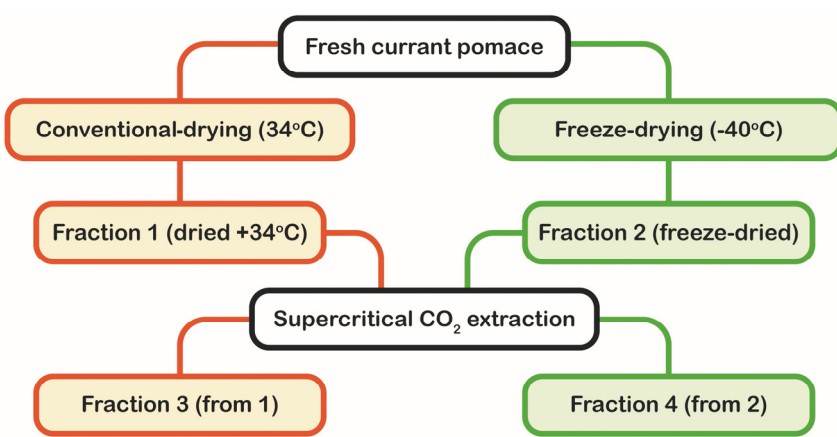

**Figure 1.** Experimental scheme of pomace drying, fractionation, and supercritical $CO_2$ extraction.

Preparation of fraction (2) for freeze-drying: Fresh blackcurrant and redcurrant pomace was spread onto lyophilisation trays and subjected to deep freezing at −80 °C for 48 h. The material was then placed in a pre-cooled freeze dryer for primary lyophilisation (set conditions: −40 °C, 0.120 mbar; achieved: −27 °C, 0.220 mbar), followed by secondary drying (set: +25 °C, 0.0010 mbar; achieved: +25 °C, 0.140 mbar), reaching 4% and 2% moisture content (R%), respectively. The dried material (2) was then used for supercritical extraction, yielding fraction (4).

Preparation of fraction (1) for conventional-drying: Fresh pomace from both currant types was distributed on drying trays and placed in a forced-air chamber dryer at +34 °C until moisture contents of 15.22% and 6.82% (R%) were reached. The dehydrated material (1) was subsequently used for supercritical fluid extraction, yielding fraction (3).

### 2.4. Supercritical Extraction Process

The dried material was volumetrically transferred to a 4 dm$^3$ extraction basket. After installation, residual air was removed using a manual vent valve. Upon pressure equalisation, extraction was initiated. The extraction time was measured from the moment the pressure reached 300 bar. The average $CO_2$ flow rate was 35.6 kg/h. After sample collection, the pressure was linearly increased to 500 bar (average flow: 99.7 kg/h). The separator pressures were maintained at 60 bar (S1) and 44 bar (S2). Depending on the trial, the extraction temperature ranged from 30 °C to 50 °C ($\pm 0.5$ °C). Samples were collected continuously without altering the pressure in the extraction basket.

### 2.5. Analytical Methods

The following analytical-grade reagents were used: gallic acid, sodium hydroxide (Sigma-Aldrich, Saint Louis, MO, USA), methanol (Honeywell, Morristown, NJ, USA), ultrapure Milli-Q water (MERCK, Burlington, MA, USA), sodium carbonate, sulfuric acid, potassium sulphate, hydrochloric acid, sulfuric acid, petroleum ether, boric acid, copper (II) sulphate and Folin–Ciocalteu reagent (all Chempur, Piekary Śląskie, Poland).

All the analytical determinations were performed for each investigated fraction of blackcurrant and redcurrant pomace to ensure comprehensive evaluation of the compositional changes resulting from the different drying and extraction conditions.

Protein determination (P) using the Kjeldahl method was performed in accordance with AOAC 920.152 [22]. The digestion phase utilised concentrated sulfuric acid ($H_2SO_4$, 95–98%), potassium sulphate ($K_2SO_4$) as a boiling point elevator, and copper (II) sulphate ($CuSO_4$) as a catalyst. For the distillation phase, 40% sodium hydroxide (NaOH) was used to liberate ammonia, which was absorbed in 4% boric acid ($H_3BO_3$) solution with mixed indicators (methyl red and methylene blue). The released ammonia was titrated with standard 0.1 N hydrochloric acid (HCl) or sulfuric acid ($H_2SO_4$).

Fat determination (F) was conducted using the Randall–Soxhlet extraction method, according to AOAC 930.09 [22] and Sójka et al. [23]. The following reagents and materials were used: petroleum ether (boiling range 40–60 °C) as the extraction solvent, pre-dried and weighed cellulose thimbles, and anhydrous sodium sulphate for sample drying prior to extraction.

The total phenolic content (TPC) was measured spectrophotometrically using the Folin–Ciocalteu method according to Sójka et al. [24] and Vorobyova et al. [25], with some modifications. Gallic acid was used as the standard, dissolved in methanol at a ratio of 10 mg GA to 5 mL MeOH. A standard curve was prepared by serial dilution, and the reaction mixture was prepared in a 2:20:1:10 ratio (standard volume/water/F–C reagent/5% $Na_2CO_3$), then brought to 50 mL in a volumetric flask and incubated for 30 min in the dark. Absorbance was measured at 760 nm.

The extracted and homogenised pomace samples (blackcurrant and redcurrant) were treated analogously and diluted in the same proportions for the TPC measurement.

### 2.6. Statistical Analysis

The experimental design was constructed using STATISTICA 13.3.721.1 software (StatSoft, Kraków, Poland) employing a Box–Behnken design (BBD) with three central points in triplicate. The independent variables were the pressure (bar), temperature (°C), and extraction time (min). The response surface methodology (RSM) was applied. The total number of experiments (N) was calculated using Formulation (1):

$$N = 2k\,(k - 1) + Cp \tag{1}$$

where k is the number of independent variables and Cp is the number of central point replications. For three variables and three replicates, the total number of experiments was 15. Table 1 below presents the predefined values of the independent variables.

**Table 1.** Codec and uncoded values of the independent variables applied in the experimental design.

| Independent Variables | Parameter | | Measured Value | |
|---|---|---|---|---|
| | Uncoded | Codec | Uncoded | Coded |
| Pressure, bar | P | $X_1$ | 300 | −1 |
| | | | 400 | 0 |
| | | | 500 | 1 |
| Extraction time, min | t | $X_2$ | 60 | −1 |
| | | | 180 | 0 |
| | | | 240 | 1 |
| Process temperature, °C | T | $X_3$ | 30 | −1 |
| | | | 40 | 0 |
| | | | 50 | 1 |

The data supporting further analysis are compiled in Tables 2–5. Table 2 outlines the baseline composition of the dried blackcurrant and redcurrant pomace (fractions 1 and 2), including the total fat, protein, and total phenolic content prior to supercritical $CO_2$ extraction.

## 3. Results

The selection of the pressure (300–500 bar) and temperature (30–50 °C) ranges was based on previously reported conditions for supercritical $CO_2$ extraction of berry seeds and fruit pomace, where pressures above 230 bar and temperatures between 35 and 45 °C were commonly applied to maximise lipid and phenolic recovery [1,2,6,21]. To provide a broader understanding of the process behaviour and evaluate the potential beyond conventional limits, the experimental design intentionally included extreme points slightly exceeding the literature-reported ranges. This approach aimed to capture possible non-linear effects of temperature on compound stability and to determine whether higher pressures could further enhance extraction yields without compromising the integrity of the thermolabile constituents.

### 3.1. Initial Composition of Dried Currant Pomace

Notably, the freeze-dried samples (fraction 2) generally exhibited higher polyphenol and protein content compared to those subjected to conventional-drying (fraction 1), while the fat content was slightly higher in the latter—reflecting the influence of the moisture removal kinetics and thermal degradation.

The chemical composition of the dried blackcurrant and redcurrant pomace prior to extraction is summarised in Table 2, which presents the content of fat (F), protein (P), and total phenolic content (TPC, mg GAE/g d.w.) for both drying methods.

**Table 2.** Fat (F), protein (P), and total phenolic content (TPC) in dried currant pomace before extraction.

| Analysis | Fraction | Parameter [Mean % d.w] |
|---|---|---|
| Soxhlet–Randall (F) | B-1-F | 12.87 ± 0.17 [a] |
| | B-2-F | 10.01 ± 0.17 [b] |
| | R-1-F | 9.41 ± 0.13 [b] |
| | R-2-F | 5.29 ± 0.10 [c] |

**Table 2.** *Cont.*

| Analysis | Fraction | Parameter [Mean % d.w] |
|---|---|---|
| Kjeldahl (P) | B-1-P | 9.43 ± 0.02 [a] |
| | B-2-P | 10.13 ± 0.05 [b] |
| | R-1-P | 8.85 ± 0.06 [c] |
| | R-2-P | 11.15 ± 0.03 [d] |

| Analysis | Fraction | Parameter [Mean mg GAE/g d.w.] |
|---|---|---|
| Folin–Ciocalteu (TPC) | B-1-TPC | 21.53 ± 2.62 [a] |
| | B-2-TPC | 31.50 ± 3.14 [b] |
| | R-1-TPC | 6.47 ± 1.12 [c] |
| | R-2-TPC | 8.33 ± 0.55 [c] |

Statistical analysis was performed using Tukey's honestly significant difference (HSD) test at a significance level of $\alpha$ = 0.05 to determine homogeneous groups for the fat content (F), protein content (P), and total phenolic content (TPC) across the experimental variants (B-1, B-2, R-1, R-2). All the analyses for fractions 1 and 2 were conducted in triplicate ($n$ = 3) to ensure reproducibility and statistical reliability. Groups that did not differ significantly were assigned the same letter, with different letters (a,b,c,d) denoting statistically distinct homogeneous groups.

### 3.2. Box–Behnken Design and Experimental Data

Tables 3–5 present the Box–Behnken design (BBD) matrices along with the experimental results for each measured response. Table 3 contains data for the total fat content, determined using the Soxhlet–Randall extraction method. Table 4 provides the total protein content based on the Kjeldahl method, and Table 5 summarises the total phenolic content assessed via the Folin–Ciocalteu assay. These datasets were used to develop the response surface models presented in Figures 2–4 (effect of pressure and temperature), Figures 5–7 (effect of pressure and time), and Figures 8–10 (effect of time and temperature) on fat, protein and polyphenol content. The experimental points are marked with blue dots, and their projections onto the response surface are shown as dashed blue lines, providing a clear visual comparison between the observed data and the model predictions.

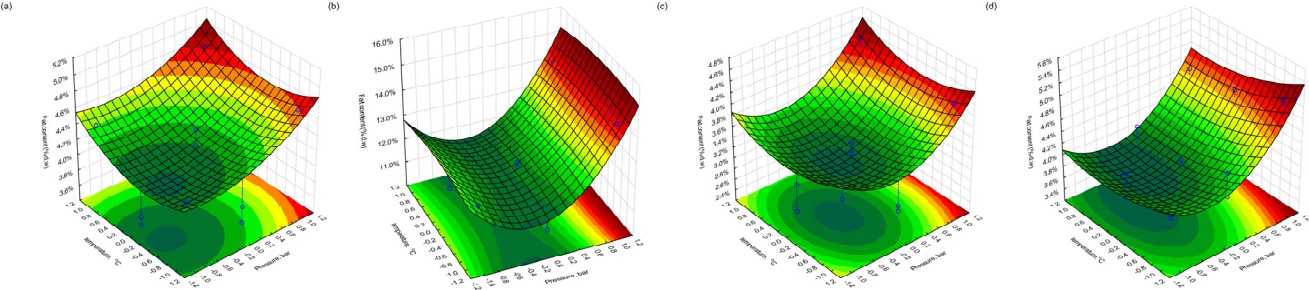

**Figure 2.** Response surface plots illustrating the influence of the extraction temperature and pressure on the total fat content (% dry weight) in currant pomace samples. Subfigure (**a**) presents the results for conventionally dried blackcurrant pomace (B-3), (**b**) for freeze-dried blackcurrant pomace (B-4), (**c**) for conventionally dried redcurrant pomace (R-3), and (**d**) for freeze-dried redcurrant pomace (R-4). Coded variables: $X_1$—pressure (bar), $X_3$—temperature (°C).

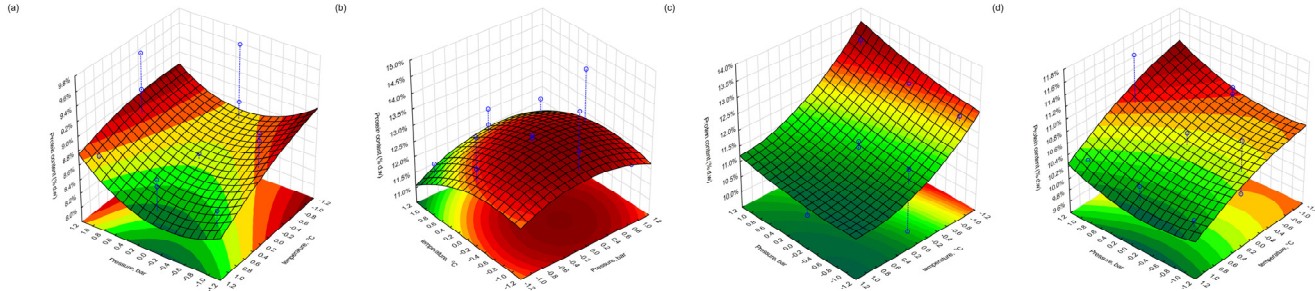

**Figure 3.** Response surface plots showing the effect of the temperature and pressure on the protein content in currant pomace: (**a**) B-3, (**b**) B-4, (**c**) R-3, and (**d**) R-4. Coded variables: $X_1$—pressure (bar), $X_3$—temperature (°C).

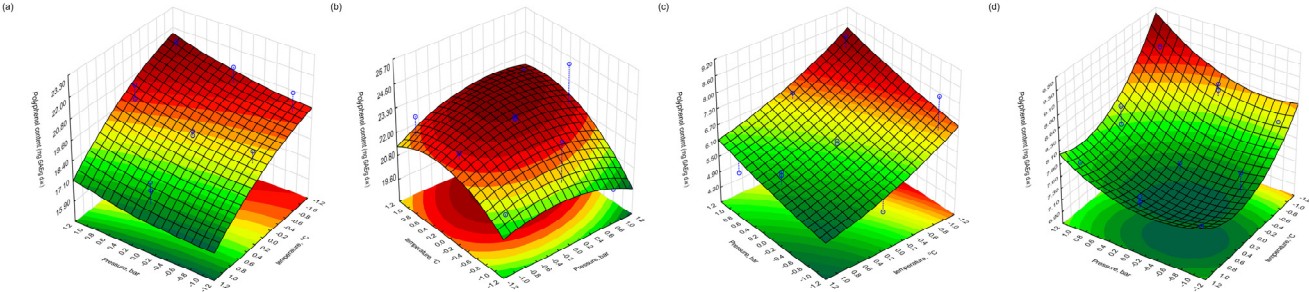

**Figure 4.** Response surface plots showing the effect of the temperature and pressure on the total phenolic content in currant pomace: (**a**) B-3, (**b**) B-4, (**c**) R-3, and (**d**) R-4. Coded variables: $X_1$—pressure (bar), $X_3$—temperature (°C).

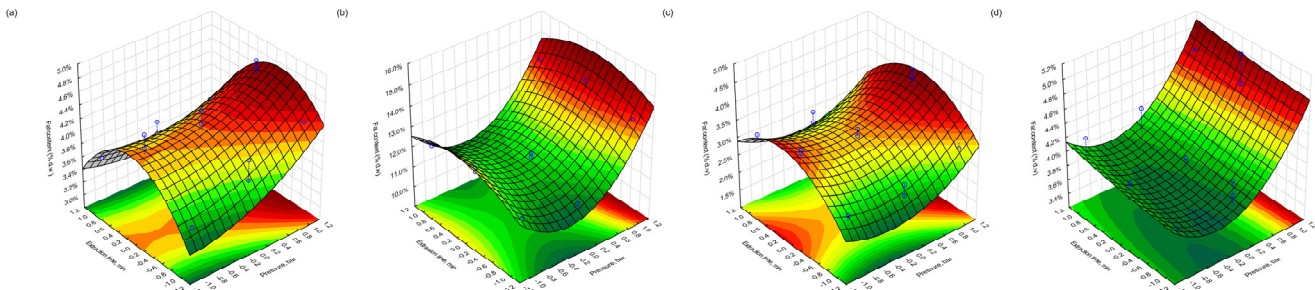

**Figure 5.** Response surface plots showing the effect of the extraction time and pressure on the total fat content (% dry weight) in currant pomace. Subfigures (**a–d**) represent the variants B-3, B-4, R-3, and R-4, respectively, corresponding to conventionally and freeze-dried samples of blackcurrant and redcurrant pomace. Coded variables: $X_1$—pressure (bar), $X_2$—time (min).

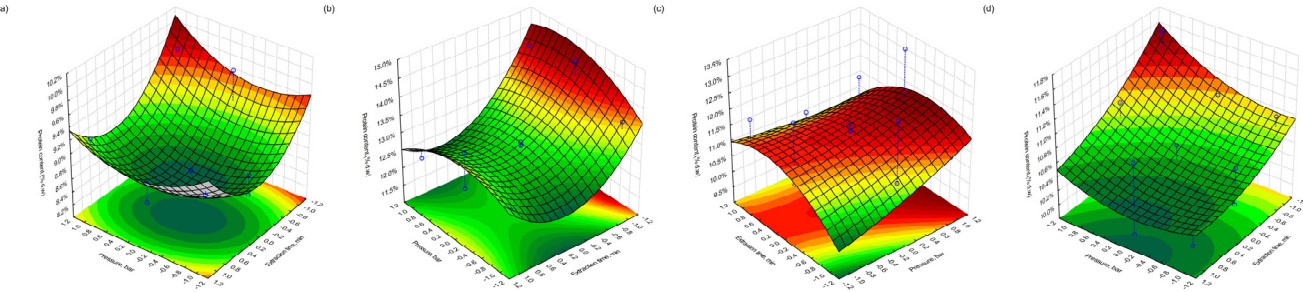

**Figure 6.** Response surface plots showing the effect of the time and pressure on the protein content in currant pomace: (**a**) B-3, (**b**) B-4, (**c**) R-3, and (**d**) R-4. Coded variables: $X_1$—pressure (bar), $X_2$—time (min).

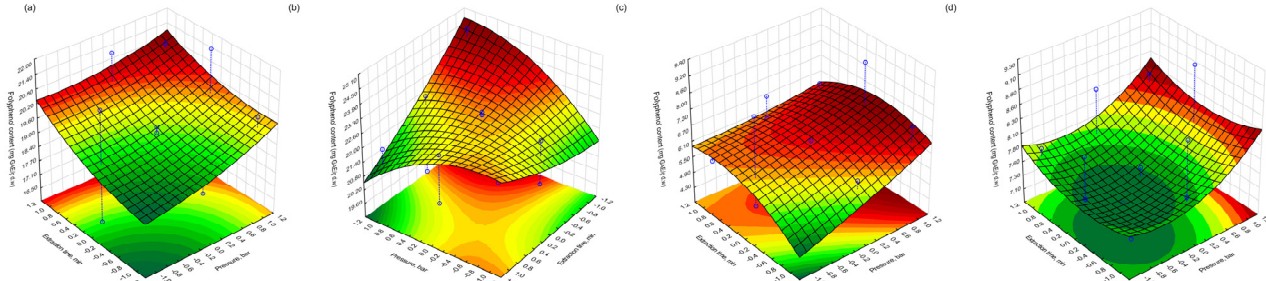

**Figure 7.** Response surface plots showing the effect of the time and pressure on the total phenolic content in currant pomace: (**a**) B-3, (**b**) B-4, (**c**) R-3, and (**d**) R-4. Coded variables: $X_1$—pressure (bar), $X_2$—time (min).

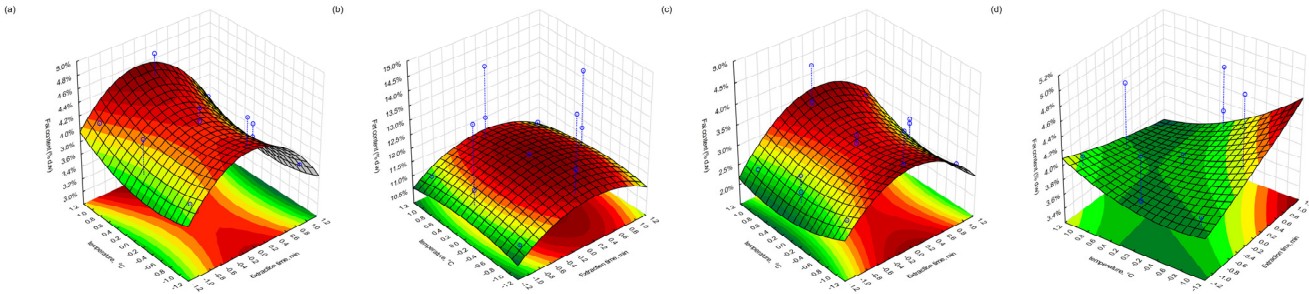

**Figure 8.** Response surface plots depicting the impact of the extraction temperature and time on the total fat content (% dry weight) in blackcurrant and redcurrant pomace. Subfigure (**a**) corresponds to B-3 (blackcurrant, conventionally dried), (**b**) to B-4 (blackcurrant, freeze-dried), (**c**) to R-3 (redcurrant, conventionally dried), and (**d**) to R-4 (redcurrant, freeze-dried). Coded variables: X2—time (min), X3—temperature (°C).

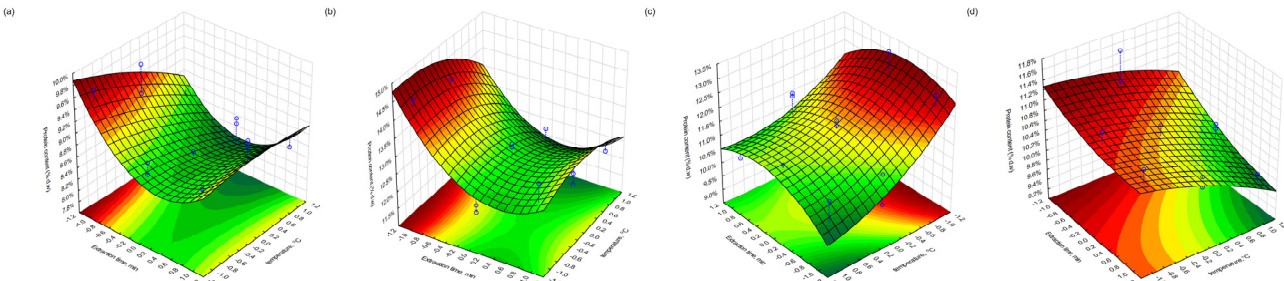

**Figure 9.** Response surface plots showing the effect of the time and temperature on the protein content in currant pomace: (**a**) B-3, (**b**) B-4, (**c**) R-3, and (**d**) R-4. Coded variables: X2—time (min), X3—temperature (°C).

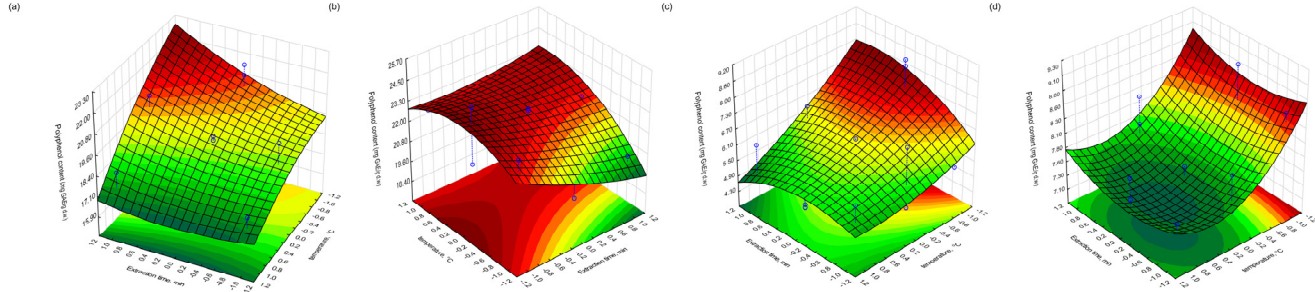

**Figure 10.** Response surface plots showing the effect of the time and temperature on the total phenolic content in currant pomace: (**a**) B-3, (**b**) B-4, (**c**) R-3, and (**d**) R-4. Coded variables: X2—time (min), X3—temperature (°C).

### 3.3. Comparative Analysis of Pre-Extraction Composition

The highest protein content was observed in freeze-dried blackcurrant pomace (fraction 2), reaching 10.1%, while the conventionally dried equivalent (fraction 1) contained 9.4%. In the case of redcurrant pomace, freeze-drying also yielded a higher protein level (11.2%) compared to the conventionally dried sample (8.9%). For the total phenolic content (TPC), the highest value was recorded in freeze-dried blackcurrant pomace (31.50 mg GAE/g d.w.), whereas the conventionally dried material contained 21.53 mg GAE/g d.w. Similarly, redcurrant pomace exhibited a higher TPC in the freeze-dried sample (8.33 mg GAE/g d.w.) than in the conventionally dried fraction (6.47 g GAE/g d.w.).

### 3.4. Influence of Drying Method and Matrix Structure

These differences are related to the heterogeneous structure of the pomace, including the varying proportions of skins, seeds, and residual stems, which differ in their chemical composition. Seeds are typically rich in lipids and proteins, while fruit skins are known to be the primary reservoir of polyphenolic compounds. Additionally, the observed variation in the polyphenol and protein content may be influenced by the drying method. Freeze-drying, conducted at low temperatures and under a vacuum, preserves thermolabile compounds more effectively and reduces oxidative degradation. In contrast, conventional-drying, despite its milder temperature (34 °C), is more time-consuming and can result in compositional loss due to prolonged exposure to oxygen and residual moisture.

Moreover, differences in the final moisture content (e.g., 4–6% for freeze-dried vs. ~15% for conventionally dried pomace) may affect the concentration effects and extraction efficiency in the later stages. The higher fat content in certain conventionally dried samples may also reflect changes in the tissue structure, enhancing the mechanical oil release, or a higher proportion of seeds in the initial material.

**Table 3.** Box–Behnken design (BBD) matrix for the efficiency of the Soxhlet fat extraction content (F) from two types of pomaces (B—*R. Nigrum*, R—*R. Rubrum*) in different processes (3–4).

| Cycle Number | Independent Variables | | | | | | Dependent Variables [% d.w] | | | |
|---|---|---|---|---|---|---|---|---|---|---|
| | X1 | P, bar | X2 | t, min | X3 | T,°C | B-3-F | B-4-F | R-3-F | R-4-F |
| 1 | −1 | 300 | −1 | 60 | 0 | 40 | 3.5 | 11.6 | 2.6 | 3.8 |
| 2 | 1 | 500 | −1 | 60 | 0 | 40 | 4.3 | 13.9 | 3.0 | 4.4 |
| 3 | −1 | 300 | 1 | 240 | 0 | 40 | 3.6 | 12.1 | 3.1 | 4.2 |
| 4 | 1 | 500 | 1 | 240 | 0 | 40 | 3.8 | 14.1 | 3.2 | 4.8 |
| 5 | −1 | 300 | 0 | 180 | −1 | 30 | 4.2 | 12.5 | 3.4 | 4.0 |
| 6 | 1 | 500 | 0 | 180 | −1 | 30 | 4.7 | 14.4 | 4.2 | 4.2 |
| 7 | −1 | 300 | 0 | 180 | 1 | 50 | 4.4 | 12.3 | 3.5 | 3.8 |
| 8 | 1 | 500 | 0 | 180 | 1 | 50 | 4.8 | 14.1 | 4.4 | 4.7 |
| 9 | 0 | 400 | −1 | 60 | −1 | 30 | 3.8 | 10.8 | 2.7 | 3.9 |
| 10 | 0 | 400 | 1 | 240 | −1 | 30 | 3.6 | 11.2 | 2.8 | 4.3 |
| 11 | 0 | 400 | −1 | 60 | 1 | 50 | 4.1 | 11.0 | 2.4 | 4.1 |
| 12 | 0 | 400 | 1 | 240 | 1 | 50 | 3.8 | 11.3 | 3.1 | 3.6 |
| 13 | 0 | 400 | 0 | 180 | 0 | 40 | 4.2 | 11.8 | 3.4 | 3.7 |
| 14 | 0 | 400 | 0 | 180 | 0 | 40 | 4.0 | 12.0 | 3.2 | 3.8 |
| 15 | 0 | 400 | 0 | 180 | 0 | 40 | 4.4 | 11.7 | 3.0 | 4.0 |

### 3.5. Residual Fat Content After Supercritical CO$_2$ Extraction

The total fat content (% d.w.) remaining in the pomace following supercritical CO$_2$ extraction was evaluated using the Soxhlet–Randall method. Among the blackcurrant

samples, the highest residual fat content was observed in fraction 4 (freeze-dried) at 14.4%, obtained under high-pressure and long-duration conditions (500 bar, 180 min, 30 °C). In contrast, the lowest fat content (3.5%) was recorded in fraction 3 (conventionally dried) following extraction at 300 bar, 60 min, and 40 °C. For redcurrant pomace, a similar trend was observed. Fraction 4 exhibited the highest fat content (5.1%) under the same intensive conditions, while fraction 3 showed the lowest residual fat (2.4%) following extraction at 400 bar, 60 min, and 50 °C.

These results suggest that the extent of the lipid removal is strongly influenced by both the drying method and the extraction parameters. The lower residual fat in conventionally dried fractions may reflect more effective extraction, possibly due to matrix porosity or thermal preconditioning that enhances solvent accessibility. Conversely, the higher fat content in the freeze-dried samples may indicate complete lipid release or structural protection of oil-containing cells, despite the use of optimised extraction conditions.

Additionally, differences in the physical structure and moisture retention of the dried matrix may have affected the mass transfer kinetics during extraction. The lower temperature (30 °C) combined with an extended extraction time and high pressure (500 bar) likely improved the $CO_2$ solvent power and selectivity for lipid-like compounds, particularly in the freeze-dried samples. However, structural resistance or aggregation of cellular components may have hindered complete lipid recovery, especially in redcurrant pomace, where the overall fat content was lower than in blackcurrant.

**Table 4.** Box–Behnken design (BBD) matrix for the efficiency of the Kjeldahl protein extraction (P) from two types of pomaces (B—*R. Nigrum*, R—*R. Rubrum*) in different processes (3–4).

| Cycle Number | Independent Variables | | | | | | Dependent Variables [% d.w] | | | |
|---|---|---|---|---|---|---|---|---|---|---|
| | **X1** | **P, bar** | **X2** | **t, min** | **X3** | **T,°C** | **B-3-P** | **B-4-P** | **R-3-P** | **R-4-P** |
| 1 | −1 | 300 | −1 | 60 | 0 | 40 | 9.2 | 13.6 | 9.7 | 11.0 |
| 2 | 1 | 500 | −1 | 60 | 0 | 40 | 9.7 | 14.2 | 10.8 | 11.6 |
| 3 | −1 | 300 | 1 | 240 | 0 | 40 | 9.1 | 12.6 | 11.7 | 10.1 |
| 4 | 1 | 500 | 1 | 240 | 0 | 40 | 9.2 | 12.3 | 11.6 | 10.3 |
| 5 | −1 | 300 | 0 | 180 | −1 | 30 | 8.9 | 11.9 | 12.5 | 10.7 |
| 6 | 1 | 500 | 0 | 180 | −1 | 30 | 8.7 | 12.1 | 13.3 | 11.0 |
| 7 | −1 | 300 | 0 | 180 | 1 | 50 | 8.6 | 11.8 | 10.6 | 10.2 |
| 8 | 1 | 500 | 0 | 180 | 1 | 50 | 8.7 | 12.4 | 11.0 | 10.3 |
| 9 | 0 | 400 | −1 | 60 | −1 | 30 | 9.7 | 14.5 | 12.7 | 11.0 |
| 10 | 0 | 400 | 1 | 240 | −1 | 30 | 8.9 | 13.3 | 11.2 | 10.9 |
| 11 | 0 | 400 | −1 | 60 | 1 | 50 | 8.7 | 12.8 | 10.8 | 10.3 |
| 12 | 0 | 400 | 1 | 240 | 1 | 50 | 8.6 | 12.3 | 10.2 | 9.9 |
| 13 | 0 | 400 | 0 | 180 | 0 | 40 | 8.5 | 12.9 | 11.5 | 10.7 |
| 14 | 0 | 400 | 0 | 180 | 0 | 40 | 8.6 | 12.5 | 11.0 | 10.4 |
| 15 | 0 | 400 | 0 | 180 | 0 | 40 | 8.5 | 12.8 | 11.7 | 10.4 |

### 3.6. Protein Content in Extracted Pomace

The total protein content (% d.w.) in the extracted pomace was determined using the Kjeldahl method. In the case of blackcurrant, the highest protein level was found in the freeze-dried sample (Fraction 4), reaching 14.5% following extraction under moderate conditions (400 bar, 60 min, 30 °C). The lowest protein content (8.5%) was observed in the conventionally dried fraction (Fraction 3), extracted at 400 bar, 180 min, and 40 °C—conditions corresponding to the central point of the experimental design. These results may indicate that prolonged extraction at moderate temperatures could lead to thermal or oxidative

degradation of nitrogenous compounds, or enhanced solubilisation and partial loss of protein-bound fractions.

In redcurrant pomace, the highest protein content (13.3%) was observed in the conventionally dried sample extracted at 500 bar, 180 min, and 30 °C, suggesting favourable conditions for retention or concentration of proteinaceous material. Conversely, the lowest value (9.7%) was found for the same drying type but under less-intensive extraction conditions (300 bar, 60 min, 40 °C). The variation in protein recovery may also reflect differences in the protein solubility under different pressure–temperature regimes, as well as structural differences in the pomace matrices caused by the drying method.

The observed differences suggest that both the drying technique and the extraction parameters influence the final protein content in the residue through their effects on protein denaturation, extractability, or matrix integrity.

**Table 5.** Box–Behnken design (BBD) matrix for the Folin–Ciocalteu total phenolic content (TPC) from two types of pomaces (B—*R. Nigrum*, R—*R. Rubrum*) in different processes (3–4).

| Cycle Number | Independent Variables | | | | | | Dependent Variables [mg GAE/g d.w] | | | |
|---|---|---|---|---|---|---|---|---|---|---|
| | X1 | P, bar | X2 | t, min | X3 | T,°C | B-3-TPC | B-4-TPC | R-3-TPC | R-4-TPC |
| 1 | −1 | 300 | −1 | 60 | 0 | 40 | 18.1 | 21.3 | 4.8 | 7.3 |
| 2 | 1 | 500 | −1 | 60 | 0 | 40 | 20.1 | 24.6 | 7.1 | 8.0 |
| 3 | −1 | 300 | 1 | 240 | 0 | 40 | 19.6 | 22.6 | 5.4 | 7.9 |
| 4 | 1 | 500 | 1 | 240 | 0 | 40 | 20.9 | 21.9 | 6.6 | 8.3 |
| 5 | −1 | 300 | 0 | 180 | −1 | 30 | 21.2 | 21.1 | 8.1 | 8.1 |
| 6 | 1 | 500 | 0 | 180 | −1 | 30 | 21.8 | 19.6 | 8.5 | 9.0 |
| 7 | −1 | 300 | 0 | 180 | 1 | 50 | 16.4 | 22.8 | 4.7 | 7.4 |
| 8 | 1 | 500 | 0 | 180 | 1 | 50 | 16.6 | 22.8 | 4.8 | 7.8 |
| 9 | 0 | 400 | −1 | 60 | −1 | 30 | 18.6 | 23.4 | 5.3 | 8.6 |
| 10 | 0 | 400 | 1 | 240 | −1 | 30 | 21.4 | 20.5 | 7.0 | 8.4 |
| 11 | 0 | 400 | −1 | 60 | 1 | 50 | 17.9 | 22.8 | 6.0 | 7.6 |
| 12 | 0 | 400 | 1 | 240 | 1 | 50 | 18.3 | 22.7 | 6.2 | 7.5 |
| 13 | 0 | 400 | 0 | 180 | 0 | 40 | 19.0 | 23.2 | 6.3 | 7.4 |
| 14 | 0 | 400 | 0 | 180 | 0 | 40 | 19.5 | 22.7 | 6.4 | 7.6 |
| 15 | 0 | 400 | 0 | 180 | 0 | 40 | 19.2 | 23.0 | 6.4 | 7.4 |

*3.7. Total Phenolic Content*

The total phenolic content (TPC, % d.w.), determined using the Folin–Ciocalteu method, showed clear dependence on both the drying method and the supercritical $CO_2$ extraction conditions. In blackcurrant pomace, the highest TPC value (24.60 mg GAE/g d.w.) was observed in the freeze-dried sample (Fraction 4) extracted at 500 bar, 60 min, and 40 °C. The lowest value (16.37 mg GAE/g d.w.) occurred under conditions of low pressure and high temperature (300 bar, 180 min, 50 °C), which may have promoted thermal degradation of polyphenolic compounds or reduced solvent selectivity.

For redcurrant pomace, the highest TPC (8.95 mg GAE/g d.w.) was also found in the freeze-dried fraction under intensive conditions (500 bar, 180 min, 30 °C), indicating the improved extraction efficiency of phenolic compounds under elevated pressure and prolonged exposure time. The lowest TPC values (4.74 mg GAE/g d.w.) were observed in the conventionally dried samples (Fraction 3), extracted under two different conditions (300 bar, 60 min, 40 °C and 300 bar, 180 min, 50 °C), further supporting the notion that the phenolic content is sensitive to both drying and extraction regimes.

The enhanced retention of phenolic compounds in the freeze-dried samples is likely due to the low-temperature dehydration process, which minimises degradation and pre-

serves thermolabile antioxidant compounds. Additionally, the higher extraction yields under specific conditions may reflect the improved solubility and diffusivity of phenolic compounds in supercritical $CO_2$, especially when combined with structurally open, porous matrices formed during freeze-drying.

### 3.8. Model Fitting and ANOVA Analysis

The experimental data presented in Tables 3–5 were used to construct second-order polynomial regression models for each of the evaluated response variables. Model fitting was performed using the response surface methodology (RSM), and statistical significance was assessed through analysis of variance (ANOVA). The quality of the model approximation was confirmed by the high coefficients of determination ($R^2$), indicating strong agreement between the predicted and observed values. Detailed regression statistics, including the $R^2$, adjusted $R^2$, and model significance levels (*p*-values), are provided in Table 6. These results validate the applicability of the RSM as a reliable tool for predicting system behaviour and optimising supercritical extraction conditions.

**Table 6.** Comparison of the post-extraction composition of blackcurrant and redcurrant pomace depending on the drying method.

| Raw Material | Drying Method | Fat (F, %) | Protein (P, %) | Total Phenolic Content (TPC, mg GAE/g d.w.) | Observed Effect |
|---|---|---|---|---|---|
| Blackcurrant | Conventional (F3) | $4.08 \pm 0.39$ | $8.90 \pm 0.39$ | $19.23 \pm 1.59$ | Higher TPC and P; highest fat recovery |
| Blackcurrant | Freeze-dried (F4) | $12.32 \pm 1.18$ | $12.80 \pm 0.77$ | $22.33 \pm 1.21$ | Highest TPC; lower content P and F |
| Redcurrant | Conventional (F3) | $3.20 \pm 0.52$ | $11.36 \pm 0.92$ | $6.25 \pm 1.09$ | Lower TPC; highest protein content |
| Redcurrant | Freeze-dried (F4) | $4.15 \pm 0.42$ | $10.59 \pm 0.44$ | $7.89 \pm 0.48$ | Highest P; reduced fat yield |

Differences may relate to structural changes in the pomace matrix, moisture content and presence of seeds or skin residues.

### 3.9. Response Surface Visualisation

The influence of the pressure, temperature, and extraction time on each response variable is presented in the form of three-dimensional response surface plots (Figures 2–10), which illustrate the interactive effects of process variables on the fat, protein, and polyphenol retention in the extracted pomace material for both drying approaches. In the plots, the blue dots denote the experimental data points, while the blue dashed lines indicate their projection onto the response surface, thereby demonstrating the correspondence between the experimental results and the model.

### 3.10. Summary of Optimal Conditions

The optimisation results obtained from the response surface methodology (RSM) indicated that the highest recovery of target components was achieved under high pressure and an extended extraction time, combined with a moderate temperature. For blackcurrant pomace, the optimal conditions predicted by the model were 500 bar, 180 min, and 30 °C, which yielded the maximum fat content (14.4%) and total phenolic content (24.60 mg GAE/g d.w.) while maintaining high protein levels. In the case of redcurrant pomace, the optimal point also corresponded to the upper pressure and time limits of the experimental range, with the slightly higher temperature (30–40 °C) contributing to the improved phenolic recovery. These results confirm that the extraction efficiency in SFE-$CO_2$ is strongly dependent on the solvent density, which increases with the pressure, and on the process duration, which influences the mass transfer and solute diffusion (Table 7).

**Table 7.** Predicted optimal SFE-CO$_2$ conditions for maximum recovery of fat (F), protein (P), and total phenolic content (TPC) from dried currant pomace.

| Raw Material | Post-Extraction Fraction | Response | Optimal Pressure (Bar) | Optimal Time (min) | Optimal Temperature (°C) | Predicted Value (%) |
|---|---|---|---|---|---|---|
| Blackcurrant | Fraction 3 | Fat (F) | 500 | 180 | 30 | 15.6 |
| Blackcurrant | Fraction 4 | Fat (F) | 500 | 180 | 30 | 14.4 |
| Redcurrant | Fraction 3 | Fat (F) | 500 | 180 | 30 | 11.2 |
| Redcurrant | Fraction 4 | Fat (F) | 500 | 180 | 30 | 5.1 |
| Blackcurrant | Fraction 3 | Protein (P) | 400 | 180 | 40 | 8.5 |
| Blackcurrant | Fraction 4 | Protein (P) | 400 | 60 | 30 | 14.5 |
| Redcurrant | Fraction 3 | Protein (P) | 500 | 180 | 30 | 13.3 |
| Redcurrant | Fraction 4 | Protein (P) | 400 | 60 | 30 | 11.2 |

| Raw Material | Post-Extraction Fraction | Response | Optimal Pressure (bar) | Optimal Time (min) | Optimal Temperature (°C) | Predicted Value (mg GAE/g) |
|---|---|---|---|---|---|---|
| Blackcurrant | Fraction 4 | Total Phenolic Content (TPC) | 500 | 60 | 40 | 24.6 |
| Redcurrant | Fraction 4 | Total Phenolic Content (TPC) | 500 | 180 | 30 | 18.6 |
| Blackcurrant | Fraction 3 * | Total Phenolic Content (TPC) | 300 | 180 | 50 | 10.2 |
| Redcurrant | Fraction 3 * | Total Phenolic Content (TPC) | 300 | 60/180 | 40/50 | 13.2 |

* Fraction 3 values for the TPC are included for comparison purposes, although the highest recoveries were obtained from the freeze-dried samples (fraction 4).

## 4. Discussion

The findings of this study demonstrate that supercritical CO$_2$ extraction (SFE-CO$_2$) enables the efficient recovery of bioactive compounds from blackcurrant and redcurrant pomace, with the extraction outcomes being strongly influenced by both the drying method and process parameters (Tables 3–5). The freeze-dried samples (fractions B-4 and R-4) consistently exhibited higher levels of total phenolic compounds (TPCs) and lipids compared to their conventionally dried counterparts (fractions B-3 and R-3), confirming the role of pre-treatment in preserving bioactive constituents. These results are in line with previous studies reporting that freeze-drying enhances antioxidant retention and lipid integrity in berry by-products [26,27]. Among the tested matrices, blackcurrant pomace showed superior extractability of all the compound classes compared to redcurrant, which can be attributed to its higher content of anthocyanins and cell-wall-associated phenolics [28,29].

Drying significantly modifies the plant matrix, thereby affecting the mass transfer and solubility during SFE. Freeze-drying generates a highly porous structure through sublimation under a vacuum, increasing the surface area and facilitating CO$_2$ penetration into intracellular spaces [14,16]. This structural openness improves access to lipophilic and moderately polar compounds embedded in the cell wall matrix, which explains the higher TPC and fat retention observed in the freeze-dried fractions (Figures 1–3). Conversely, hot-air-drying often induces tissue shrinkage and partial cell collapse due to thermal and mechanical stress, resulting in denser matrices with lower porosity and higher diffusion resistance [15]. The residual moisture in conventionally dried pomace can further hinder CO$_2$ transport or alter its solvating power by promoting localised swelling, whereas freeze-dried materials, with minimal water content, ensure more consistent solvent interaction [17]. These structural differences highlight the importance of the drying method selection as a critical factor influencing supercritical extraction performance.

The influence of individual process variables was evident across all the responses. Elevated pressure and an extended extraction time favoured the recovery of lipids and phenolic compounds, whereas the protein content exhibited greater sensitivity to tem-

perature (Figures 4–9). This behaviour may reflect changes in solubility under varying supercritical conditions and the susceptibility of nitrogenous compounds to thermal degradation [27]. The highest fat content in the post-extraction matrix (14.4%) and maximum TPC (24.60 mg GAE/g d.w.) were obtained under high pressure (500 bar) and prolonged extraction (180 min) in the freeze-dried blackcurrant samples, underscoring the matrix- and parameter-specific nature of the process (Table 7). These findings align with previous research indicating that SFE performance is optimised when the solvent density and exposure time are sufficient to overcome the mass transfer limitations [29,30].

Previous studies have reported promising results for the application of supercritical $CO_2$ extraction in recovering oils and phenolic compounds from berry seeds and pomace; however, most investigations focused on individual parameters or small-scale systems [2,6,10,19]. Research typically emphasises specific aspects such as the lipid yield from blackcurrant seeds [10], phenolic retention under alternative extraction techniques [7,8], or the effects of single process variables on recovery efficiency [1,4]. In contrast, the present work proposes an integrated model incorporating multiple variables, including the pre-treatment conditions (freeze-drying vs. conventional-drying), the heterogeneity of unclassified raw pomace, and scale-dependent equipment characteristics. These factors introduce complexity that makes direct comparison with previously published datasets challenging and, in many cases, not fully objective. Consequently, the literature benchmarks can only serve as an indicative reference for validating the rationale behind combining drying pre-treatments with SFE, rather than as a basis for strict quantitative alignment. This approach reflects the practical diversity encountered in industrial processing of berry residues and highlights the importance of multifactorial optimisation strategies for process intensification.

The compositional profile of the extracts obtained under optimised SFE conditions—rich in PUFA and phenolic antioxidants—suggests potential applications in functional food, nutraceutical, and cosmetic formulations. Previous reports have demonstrated that berry-derived extracts improve oxidative stability and confer health-promoting benefits when incorporated into meat products, emulsions, and dietary supplements [26]. Compared to conventional solvent-based or ultrasound-assisted extractions, SFE offers the advantages of tuneable selectivity, solvent-free operation, and product purity, which are crucial for food-grade applications [27–29]. Furthermore, coupling SFE with downstream processes such as encapsulation or fractionation could enhance the stability and usability of bioactive compounds in industrial practice. These results reinforce the view that supercritical $CO_2$ extraction is not only a highly efficient recovery technique but also a scalable and environmentally responsible strategy for closing material loops in the fruit processing sector [30].

Recent studies further support the importance of integrating green extraction technologies within circular economy frameworks. Kandemir et al. (2022) emphasised that fruit juice industry by-products represent an abundant source of polyphenols, flavonoids, and dietary fibre suitable for use in food, cosmetic, and pharmaceutical formulations [31]. Similarly, Fomo et al. (2020) identified SFE as a key enabler of the sustainable valorisation of agro-industrial waste, citing its ability to reduce solvent use and deliver high-purity extracts [32]. The outcomes of this study confirm these assertions by demonstrating that SC-$CO_2$ extraction applied to properly pre-treated pomace ensures efficient recovery of thermolabile and lipophilic constituents while supporting sustainability-driven product innovation.

Despite its numerous advantages, supercritical $CO_2$ extraction (SFE) also presents several limitations that must be considered when implementing this technology at an industrial scale. The high capital investment associated with pressure-resistant equipment and the need for precise process control significantly increase the operational costs com-

pared to conventional extraction methods [1,2,4]. Additionally, the energy demand for maintaining high pressures (typically above 200 bar) and temperature stability contributes to the overall process costs and environmental footprint, despite the solvent-free nature of the method [1,5]. Process efficiency can also be affected by the low polarity of $CO_2$, which limits its ability to solubilise highly polar bioactive compounds such as phenolic acids unless co-solvents (modifiers) like ethanol or water are added [2,3,6]. The necessity of incorporating these modifiers introduces additional steps for solvent removal and quality assurance, potentially impacting the "green" character of the process [3,4]. Furthermore, scaling up SFE for continuous operation presents technical challenges related to the extraction kinetics, solid matrix heterogeneity, and mass transfer limitations in industrial-scale reactors [1,4,6]. These factors highlight that, although SFE offers a sustainable alternative to conventional solvent-based extractions, its widespread adoption in the food and nutraceutical industries will require further optimisation of the equipment design, process integration, and cost-reduction strategies to ensure economic feasibility [2,4,5].

The experimental design did not include the addition of polar co-solvents during supercritical $CO_2$ extraction, as this study aimed to standardise the analytical outcomes across all the tested responses rather than to maximise the recovery of a single compound class such as phenolics. This approach allowed the process evaluation to focus on the influence of the pressure, temperature, and extraction time—variables that were systematically controlled according to the Box–Behnken design—without introducing confounding effects from solvent polarity modifiers. Moreover, given the energy-intensive nature of SFE and the use of high-pressure equipment, omitting co-solvents minimised the operational complexity and ensured that the observed differences in the extract composition could be attributed primarily to the selected process parameters rather than additional chemical interactions.

The choice of freeze-drying and low-temperature conventional-drying as pre-treatment methods was justified by their contrasting impact on the matrix structure and bioactive compound stability. Freeze-drying was selected due to its well-documented ability to preserve thermolabile compounds, such as phenolics and ascorbic acid, by limiting oxidative and thermal degradation during dehydration. In contrast, conventional-hot-air-drying, although less protective, reflects an economically relevant method commonly applied in the food industry for processing large volumes of pomace. Including both techniques allowed for a comparative assessment of two widely different scenarios—one prioritising quality retention and the other representing practical industrial feasibility—thereby providing a comprehensive evaluation of their effect on the supercritical $CO_2$ extraction efficiency when applied to berry-derived by-products.

## 5. Conclusions

This study demonstrates that supercritical $CO_2$ extraction is an effective method for obtaining lipid- and phenolic-compound-rich extracts from blackcurrant and redcurrant pomace. The results indicate that both the drying method and the extraction parameters significantly affect the yield and composition of the recovered bioactive compounds. The freeze-dried samples showed superior extraction efficiency, particularly in terms of the total phenolic compounds and fat content, highlighting the importance of pre-treatment in valorising agro-industrial residues. Among the tested variables, the pressure and time had the most pronounced effects on the extraction performance. These findings support the potential of using optimised SFE processes to valorise fruit pomace as a functional ingredient source in food, cosmetic, and nutraceutical applications. Future work will focus on evaluating the antioxidant activity of the obtained extracts using DPPH and ABTS assays to better understand their functional potential and applicability in formulation development.

**Author Contributions:** Conceptualisation, F.H. and M.K.; methodology, F.H. and M.K.; formal analysis, F.H.; data curation, F.H.; writing—original draft preparation, F.H. and M.K.; writing—review and editing, F.H.; visualisation, M.K.; supervision, M.K. All authors have read and agreed to the published version of the manuscript.

**Funding:** This research received no external funding.

**Institutional Review Board Statement:** Not applicable.

**Informed Consent Statement:** Not applicable.

**Acknowledgments:** The fifth edition of the implementation doctorate programme—Ministry of Science and Higher Education.

**Conflicts of Interest:** The authors declare no conflicts of interest.

## Abbreviations

The following abbreviations are used in this manuscript:

| | |
|---|---|
| TPC | Total Phenolic Content—Folin–Ciocalteu method |
| F | Fat Content—Soxhlet–Randall method |
| P | Protein Content—Kjeldahl method |
| SC-$CO_2$ | Supercritical Carbon Dioxide |
| SFE | Supercritical Fluid Extraction |
| SFE-$CO_2$ | Superfluid Extraction Carbon Dioxide |
| PUFA | Polyunsaturated Fatty Acid |
| d.w. | Dry Weight |

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
