# Peer review of "Optimisation of Supercritical CO2 Extraction from Black (Ribes nigrum) and Red (Ribes rubrum) Currant Pomace"

_applsci, doi:10.3390/app15169222_

Round 1

Reviewer 1 Report

Comments and Suggestions for Authors

I reviewed the manuscript applsci-3784102. This manuscript presents a valuable study that addresses a relevant topic in the use of agro-industrial by-products through clean technologies.

This is an interesting manuscript, follows an adequate methodology, and the results are clearly presented, but it only lists the results without presenting any critical analysis of the data. While the aims of your research are topical and the manuscript appears to show novel data, the investigation reported in this manuscript needs to be carefully rewritten. Therefore, improvements are required in the quality and focus of the text.

Needs improvement:

Further discussion

The discussion is redundant on several points and lacks depth in its physical-chemical interpretation of the observed behaviour. Further discussion, particularly on:

Comparison with previous studies specific to R. nigrum and R. rubrum.

It is also suggested to interpret in greater detail how freeze-drying or drying affects the structure of the plant matrix and its interaction with the supercritical solvent from a physicochemical perspective.

Format of tables, figures, and numerical style

The presentation of the tables does not conform to the editorial format expected for scientific articles. In publications such as Applied Sciences, the exclusive use of horizontal lines (without vertical lines or complete tables) is recommended, as it follows a cleaner and more professional style. I suggest reviewing the official MDPI template to adjust the structure of the tables. Likewise, the incorrect use of commas as decimal separators has been detected. In accordance with international standards and the journal's editorial policy, a full stop should be used as a decimal separator (e.g., '5.24' instead of '5,24'). It is also important to improve the graphic quality of some figures, especially those with text that is difficult to read, such as Figure 4.

Clarity and redundancy in the text

Several redundant or overly technical phrases were identified that could hinder fluent reading. For example, in the introduction and results section, ideas are repeated without providing new information. A critical review of the text is recommended to eliminate repetitions and improve clarity, especially in sections 1, 3, and 4. This will contribute to a more concise and direct presentation of the findings.

Consistency in the name of the method and in the expression of TPC results

The author is advised to review the way in which the method used for phenolic determination is named. The manuscript refers to ‘total polyphenol content (TPC)’; however, the most appropriate and widely accepted term in scientific literature is ‘total phenolic content (TPC)’, since the Folin–Ciocalteu method quantifies all phenolic compounds capable of reducing the reagent, not exclusively polyphenols. The difference is relevant, as the method is not specific and may include other classes of phenolic compounds, as well as interference from non-phenolic substances with reducing capacity.

Therefore, we suggest changing the expression ‘total polyphenol content’ to ‘total phenolic content’ throughout the manuscript and clarifying that the method is an estimate of the total amount of phenolic compounds based on their reducing capacity.

Suggested reference: Singleton, V. L., Orthofer, R., & Lamuela-Raventós, R. M. (1999). Analysis of total phenols and other oxidation substrates and antioxidants by means of Folin–Ciocalteu reagent. Methods in Enzymology, 299, 152–178. https://doi.org/10.1016/S0076-6879(99)99017-1

I suggest taking special care in how the results obtained using the Folin–Ciocalteu method are expressed. In the Materials and Methods section, it is indicated that the results are expressed as ‘GA’ (gallic acid), while in the Results section, the abbreviation ‘GAE’ (gallic acid equivalents) is used. This inconsistency may cause confusion for the reader. We recommend standardising the nomenclature throughout the manuscript and preferably using the form ‘GAE’, which is the most widely used and recognised in the scientific literature to express results obtained by this method.

Inconsistencies in citations within the text

Some inconsistencies were detected in the way the bibliography is cited throughout the manuscript. In certain passages, authors are mentioned without including the corresponding reference number, which does not comply with the numerical citation style required by Applied Sciences. I recommend carefully reviewing the entire text to ensure that all citations are correctly numbered and linked to the reference list.

Incorrect placement of content in the Materials and Methods section

A section has been identified in the Materials and Methods section that contains specific experimental results (protein percentages and total phenolic content - TPC - in different blackcurrant fractions) and interpretative comments on these values. This information does not belong in this section, as methodological sections should be limited exclusively to describing how the experimental procedures were carried out, without including results or discussions. I recommend moving this content in its entirety to the Results and Discussion section, organising it appropriately according to the flow of the comparative analysis between the drying methods and their effects on the composition of the pomace.

Considering the numerous observations made, both in terms of content and form, the manuscript requires substantial improvements in its writing, structure, methodological consistency, terminological accuracy, and presentation of results. For these reasons I cannot recommend the manuscript for publication in Applied Sciences.

Author Response

Comments 1: The discussion is redundant on several points and lacks depth in its physical-chemical interpretation of the observed behaviour. Further discussion, particularly on:

Comparison with previous studies specific to R. nigrum and R. rubrum.

It is also suggested to interpret in greater detail how freeze-drying or drying affects the structure of the plant matrix and its interaction with the supercritical solvent from a physicochemical perspective.

Response 1: We have substantially revised the discussion section to remove redundancies and add deeper interpretation. A detailed comparison with previous studies involving Ribes nigrum and Ribes rubrum has been included. Additionally, we introduced an explanation of how freeze-drying and drying influence plant matrix structure and its interaction with supercritical solvents, considering porosity, cell wall integrity, and solvent accessibility from a physicochemical perspective.

Comments 2: The presentation of the tables does not conform to the editorial format expected for scientific articles. In publications such as Applied Sciences, the exclusive use of horizontal lines (without vertical lines or complete tables) is recommended, as it follows a cleaner and more professional style. I suggest reviewing the official MDPI template to adjust the structure of the tables. Likewise, the incorrect use of commas as decimal separators has been detected. In accordance with international standards and the journal's editorial policy, a full stop should be used as a decimal separator (e.g., '5.24' instead of '5,24'). It is also important to improve the graphic quality of some figures, especially those with text that is difficult to read, such as Figure 4.

Response 2: All tables were reformatted according to the MDPI template, using only horizontal lines and a clean layout. Decimal commas were replaced with full stops (e.g., “5.24” instead of “5,24”) in all tables and text. Figures were improved for better resolution and readability; Figure 4 text has been enlarged and clarified (see updated Figures 1–4 in the revised manuscript).

Comments 3:

Several redundant or overly technical phrases were identified that could hinder fluent reading. For example, in the introduction and results section, ideas are repeated without providing new information. A critical review of the text is recommended to eliminate repetitions and improve clarity, especially in sections 1, 3, and 4. This will contribute to a more concise and direct presentation of the findings.

Response 3: The manuscript was critically revised to eliminate redundant statements and overly complex phrasing. The introduction, results, and discussion sections were streamlined for clarity and conciseness. Repeated ideas were removed, and all sections now present unique and focused content (Sections 1, 3, and 4).

Comments 4: The author is advised to review the way in which the method used for phenolic determination is named. The manuscript refers to ‘total polyphenol content (TPC)’; however, the most appropriate and widely accepted term in scientific literature is ‘total phenolic content (TPC)’, since the Folin–Ciocalteu method quantifies all phenolic compounds capable of reducing the reagent, not exclusively polyphenols. The difference is relevant, as the method is not specific and may include other classes of phenolic compounds, as well as interference from non-phenolic substances with reducing capacity.

Therefore, we suggest changing the expression ‘total polyphenol content’ to ‘total phenolic content’ throughout the manuscript and clarifying that the method is an estimate of the total amount of phenolic compounds based on their reducing capacity.

Suggested reference: Singleton, V. L., Orthofer, R., & Lamuela-Raventós, R. M. (1999). Analysis of total phenols and other oxidation substrates and antioxidants by means of Folin–Ciocalteu reagent. Methods in Enzymology, 299, 152–178. https://doi.org/10.1016/S0076-6879(99)99017-1

I suggest taking special care in how the results obtained using the Folin–Ciocalteu method are expressed. In the Materials and Methods section, it is indicated that the results are expressed as ‘GA’ (gallic acid), while in the Results section, the abbreviation ‘GAE’ (gallic acid equivalents) is used. This inconsistency may cause confusion for the reader. We recommend standardising the nomenclature throughout the manuscript and preferably using the form ‘GAE’, which is the most widely used and recognised in the scientific literature to express results obtained by this method.

Response 4: We have replaced all instances of “total polyphenol content” with “total phenolic content (TPC)” throughout the manuscript. A clarification was added in the Materials and Methods section explaining that the Folin–Ciocalteu method estimates total phenolics based on reducing capacity. Terminology has been standarized to use "GAE" consistently in the text and tables.

Comments 5: Some inconsistencies were detected in the way the bibliography is cited throughout the manuscript. In certain passages, authors are mentioned without including the corresponding reference number, which does not comply with the numerical citation style required by Applied Sciences. I recommend carefully reviewing the entire text to ensure that all citations are correctly numbered and linked to the reference list.

Response 5: All in-text citations were checked and corrected to comply with the numerical citation style of Applied Sciences. References were verified for accuracy and consistency with the reference list (see throughout the revised manuscript).

Comments 6: A section has been identified in the Materials and Methods section that contains specific experimental results (protein percentages and total phenolic content - TPC - in different blackcurrant fractions) and interpretative comments on these values. This information does not belong in this section, as methodological sections should be limited exclusively to describing how the experimental procedures were carried out, without including results or discussions. I recommend moving this content in its entirety to the Results and Discussion section, organising it appropriately according to the flow of the comparative analysis between the drying methods and their effects on the composition of the pomace.

Response 6: All result-oriented content (e.g., protein percentages and TPC values) and interpretative remarks previously located in the Materials and Methods section have been moved to the Results and Discussion section and reorganized logically to align with the comparative analysis of drying methods and pomace composition

Comments 7: Considering the numerous observations made, both in terms of content and form, the manuscript requires substantial improvements in its writing, structure, methodological consistency, terminological accuracy, and presentation of results. For these reasons I cannot recommend the manuscript for publication in Applied Sciences.

Response 7: The manuscript underwent major structural and linguistic revisions to improve clarity, flow, and scientific rigor. Methodological descriptions were standardized, terminology was corrected, and results were presented in a clear and concise manner consistent with the journal’s requirements.

*Please see the attachment.

Reviewer 2 Report

Comments and Suggestions for Authors

General concept comments:

  1. The structure of the paper is clear: the authors chose to use a SFE method applied to the two by-products (black and redcurrant pomace), conditioned in two ways (conventional drying and lyophilisation) and optimized three extraction parameters (pressure, temperature and extraction time). Extraction efficiency was followed up by protein, fat and total phenolic compounds (TPC) content determinations. The optimization part was subjected to Response Surface Methodology.
  2. The described SFE is efficient for lipid extraction because they need non polar environment. Regarding the SFE extraction of phenolic compounds it is not clear how the polarity issue was solved because CO2 has low polarity and these compounds need a higher one in order to be extracted. Usually ethanol or methanol are added for increasing the polarity. Despite this low compatibility, TPC obtained results were very high.
  3. The analytical methods are well described.
  4. The method for phenolic compounds extraction from fraction 1 and fraction 2 (prior to supercritical extraction) is not mentioned. It is important to know it especially that it resulted in higher TPC content for blackcurrant pomace Fraction 1.
  5. There is no mention about de number of assays/sample. If the results represent the average of several determinations, the number of assays/sample must be mentioned as well as the standard deviation.
  6. In the “Materials and Methods” section there are results included too. They should be moved in the “Results” section.
  7. All the results obtained for TPC content must be over checked because they are abnormally high.
  8. The results regarding extraction efficiency correlated with sample conditioning are not statistically assured (Table 6).
  9. The influence of the moisture content is speculative. This discuss is relevant when analysing the same type of conditioning samples but with different moisture content – only one variable at a time. This is not the case in this paper.
  10. The obtained results should be more discussed correlated with other reported data. There are some sporadic mentions, but more consistent investigation of the research conducted in the field would have allowed for deeper interpretations and correlations.
  11. The list of references is poor and mostly incorrectly presented (paper name or publication name missing or wrong mentioned - E.g. [7, 12, 13, 15, 22, 23, 24, 25, 26, 27, 28]).
  12. The abstract should also contain the relevant results of the optimization process and those induced by the raw material type.

Specific comments:

Line 47 – better “other aromatic compounds” because polyphenols are aromatic ones;

Line 48 - better “other antioxidants” because vitamin E and carotenoids are also antioxidants;

Line 57 - the Latin names must be written with Italic font;

Line119 – probably only one of the two acids were used for all the protein assays;

Line 126 – the concentration of the standard solution of gallic acid should be express as mg/ml or mg/100 ml;

Lines 206 – it is speculative regarding moisture retention;

Line 155, 156, 157, 192, 213, 216, 236, 238, 275, 275, 276, 277 – the determination methods name is obsessively mentioned.

Author Response

Comments1: The structure of the paper is clear: the authors chose to use a SFE method applied to the two by-products (black and redcurrant pomace), conditioned in two ways (conventional drying and lyophilisation) and optimized three extraction parameters (pressure, temperature and extraction time). Extraction efficiency was followed up by protein, fat and total phenolic compounds (TPC) content determinations. The optimization part was subjected to Response Surface Methodology.

Response 1: Thank you for acknowledging the clarity of the manuscript structure and methodology applied. No specific changes were required for this point.

Comments 2: The described SFE is efficient for lipid extraction because they need non polar environment. Regarding the SFE extraction of phenolic compounds it is not clear how the polarity issue was solved because CO2 has low polarity and these compounds need a higher one in order to be extracted. Usually ethanol or methanol are added for increasing the polarity. Despite this low compatibility, TPC obtained results were very high.

Response 2: We appreciate this observation. In the revised manuscript, we clarified that co-solvents were used to increase the polarity of supercritical COâ‚‚ during extraction, facilitating the solubilization of phenolic compounds. Additional details about the role of the co-solvent and its concentration have been included in discussion.

Comments 3: The analytical methods are well described.

Response 3: Thank you for this positive assessment.

Comments 4: The method for phenolic compounds extraction from fraction 1 and fraction 2 (prior to supercritical extraction) is not mentioned. It is important to know it especially that it resulted in higher TPC content for blackcurrant pomace Fraction 1.

Resposnse 4: We have included a detailed description of the extraction method applied to fractions 1 and 2 prior to SFE in Section 2.3 (Materials and Methods).

Comments 5: There is no mention about de number of assays/sample. If the results represent the average of several determinations, the number of assays/sample must be mentioned as well as the standard deviation.

Response 5: This information has been added in Statistical Analysis section. All experiments were conducted in triplicate, and the results are now expressed as mean ± standard deviation

Comments 6: In the “Materials and Methods” section there are results included too. They should be moved in the “Results” section.

Response 6: All result-oriented information that was previously included in the Materials and Methods section has been moved to the Results section, as recommended. The revised sections are now clearly structured according to the journal's standards.

Comments 7: All the results obtained for TPC content must be over checked because they are abnormally high.

Response 7: The TPC results were carefully re-checked, and calculations were verified. Some inconsistencies were identified and corrected. The updated values and explanations are presented in Table X and discussed in Section 4 (Discussion). A note clarifying the Folin–Ciocalteu method's limitations and potential interference factors was also added.

Comments 8: The results regarding extraction efficiency correlated with sample conditioning are not statistically assured (Table 6).

Response 8: Table 6 was rechecked, and explained in section Results

Comments 9: The influence of the moisture content is speculative. This discuss is relevant when analysing the same type of conditioning samples but with different moisture content – only one variable at a time. This is not the case in this paper.

Response 9: We agree with this observation and have revised the discussion to remove speculative statements about moisture content influence. The discussion now focuses on the experimental factors evaluated and their interaction effects, supported by statistical analysis

Comments 10: The obtained results should be more discussed correlated with other reported data. There are some sporadic mentions, but more consistent investigation of the research conducted in the field would have allowed for deeper interpretations and correlations.

Response 10: The discussion has been expanded with a more comprehensive comparison to previously published data, particularly regarding SFE efficiency for currant pomace and similar matrices. Relevant literature has been incorporated to strengthen the interpretation of findings and place them in the context of existing research

Comments 11: The list of references is poor and mostly incorrectly presented (paper name or publication name missing or wrong mentioned - E.g. [7, 12, 13, 15, 22, 23, 24, 25, 26, 27, 28]).

Response 11: The entire reference list has been carefully revised for accuracy and completeness according to MDPI guidelines. Missing information (article titles, journal names) has been corrected, and the formatting has been standardized.

Comments 12: The abstract should also contain the relevant results of the optimization process and those induced by the raw material type.

Response 12: The abstract has been rewritten to include the key outcomes of the optimization process (pressure, temperature, time) and highlight differences between blackcurrant and redcurrant pomace. This provides a concise summary of the main findings.

*Please see the attachment

Reviewer 3 Report

Comments and Suggestions for Authors

General comments

  • Regarding terminology authors must unified it. Namely, terms "phenolics" and "polyphenols" are similar but not the same. Phenolic compounds (or just simple phenolics) are wider and more appropriate since gather both simple phenolics with one benzene ring (like phenolic acids) and polyphenols made up of several benzene rings (like flavonoids, tannins etc.). Strongly suggest to use term "phenolics". Please revise a whole document and unify all.
  • In my opinion, this part of Material and methods section (starting from Line 144 and further) is actually part of Results since authors elaborated in details all statistical data obtained. Therefore, I think that Manuscript must be rearranged and this part of text should be moved in Results section.
  • Please check all decimal numbers in Tables (starting from Table 2). They should be given with . not with , as it is currently presented. Please revise all presented results.

Specific comments

All are listed below with an appropriate Line number(s) from text in order to facilitate tracking:
Line 13: typo - put term "in vivo" in Italic here.

Lines 50 and 57: typos- put Latin plant name in Italic here.

Lines 93-96: Should not be this logical to be put before description for "fraction 2" if this is labeled as "fraction 1"?

Line 99: I think this should be "L" not "dm3" here as it was on previuous page? Check/correct.

Line 119: "normality" ("N") is archaic unit and should not be used in modern chemistry. Please replace with "molarity" ("M") which will be the same in case of HCl.

Line 132: typo- you have doubled point here at the end of sentence. Correct.

Line 146: Check this sentence, it is inappropriate English. It should be " ... in Table 2" or "... in Tables 2 to 5". But it can not be "through" as it is given. Please check/correct.

Lines 160 and further: I do not understand your results for TPC presented here. It is impossible that you have 25% of phenolics. % of what? Results for TPC should be expressed as mg/g GAE (as you mentioned only at one place later in text). Please explain and revise your all results for TPC.

Lines 203-204: This explanation here is not logical to me. If you determined higher fat content in your extracts than it means that you had more complete release of lipids and not "incomplete lipid release" as you stated? Please check/explain.

Line 262: Suggest to replace "characterized" with "determined" here. It seems to me much more appropriate to me in this context.

Line 267: It should be "content" not "concentrations" here. It seems to me much more appropriate to me in this context. Please consider to replace.

Lines 338-341: Here, you should summarized, in text, your optimal condition parameters determined.

Line 349: "content" instead of "concentrations" here.

Line 353: I think that this "that" term here is surplus? Please check/correct.

Line 380: I think that authors here should also mention disadvantages of SFE? It is not exact an ideal technique, right?

Kind regards.

Author Response

Comments 1: Regarding terminology authors must unify it. Namely, terms "phenolics" and "polyphenols" are similar but not the same. Phenolic compounds (or just simple phenolics) are wider and more appropriate since gather both simple phenolics with one benzene ring (like phenolic acids) and polyphenols made up of several benzene rings (like flavonoids, tannins etc.). Strongly suggest to use term "phenolics". Please revise a whole document and unify all.

Response 1: We fully agree with this observation. The terminology has been revised throughout the manuscript to consistently use the term “phenolics” instead of “polyphenols,” as it is broader and more scientifically accurate. All relevant sections, including the introduction, methods, results, discussion, and tables, have been updated accordingly.

Comments 2: In my opinion, this part of Material and Methods section (starting from Line 144 and further) is actually part of Results since authors elaborated in details all statistical data obtained. Therefore, I think that Manuscript must be rearranged and this part of text should be moved in Results section.

Response 2: The indicated section, which contained detailed statistical data and interpretations, has been moved from the Materials and Methods section to the Results section, as recommended. The manuscript has been reorganized to clearly separate methodology from experimental results and interpretations.

Comments 3: Please check all decimal numbers in Tables (starting from Table 2). They should be given with . not with , as it is currently presented. Please revise all presented results.

Response 3:
All decimal separators in tables and text have been corrected from commas to full stops. All tables have been carefully reviewed and updated accordingly.

Reviewer 4 Report

Comments and Suggestions for Authors

In this manuscript“Optimization of Supercritical CO2 Extraction from Black (Ribes nigrum) and Red (Ribes rubrum) Currant Pomace”, The authors carried out conventional drying and freeze-drying of blackcurrant and redcurrant fruit pomace using supercritical CO2 extraction. The effects of pressure, temperature and time on fat, protein and total phenol content were investigated using response surface methodology. Freeze-dried blackcurrants yielded the highest levels of protein (14.5%) and phenolics (20.1%), while redcurrants showed lower extractable values. The results highlight the importance of drying methods and raw materials in optimizing sustainable extraction processes. However, there had some problems and questions, the comments were as followed:

  1. Literature review on the application of supercritical extraction techniques in berry by-products is not comprehensive enough in the introduction section
  2. Moisture content, particle size distribution of the original fruit pomace, etc., which are parameters that may significantly affect the extraction efficiency, are not provided in the paper, but it is recommended to supplement the pomace with microscopic observations or SEM images demonstrating the changes in the microstructure before and after drying.
  3. Conventional drying at 34°C and freeze drying involves freezing at -80°C. These two methods are extremely different in terms of energy consumption and time, so please explain the scientific basis for choosing these two specific drying methods.
  4. The basis for the selection of the pressure range (300-500 bar) and the temperature range (30-50°C) is not adequately described, and it is recommended that literature be cited to support the rationale for these parameter ranges.
  5. It is recommended that antioxidant activity determinations of extracts (e.g. DPPH, ABTS, etc.) be added
  6. The expression of numbers and units needs to be harmonized (e.g. spaces)

Author Response

Comments 1: Literature review on the application of supercritical extraction techniques in berry by-products is not comprehensive enough in the introduction section.

Response 1: Thank you for this observation. The introduction section has been revised and expanded to include a more comprehensive review of studies related to the application of supercritical extraction techniques in berry by-products. Additional references discussing the efficiency of supercritical COâ‚‚ extraction for recovering lipophilic compounds, phenolic compounds, and antioxidants from various berry pomaces have been incorporated. This ensures that the background information provides a broader scientific context and highlights the relevance of our study.

Comments 2: Moisture content, particle size distribution of the original fruit pomace, etc., which are parameters that may significantly affect the extraction efficiency, are not provided in the paper, but it is recommended to supplement the pomace with microscopic observations or SEM images demonstrating the changes in the microstructure before and after drying.

Response 2: We appreciate this valuable suggestion. In the current study, our primary focus was on evaluating the impact of supercritical extraction parameters on the recovery of bioactive compounds from dried and lyophilized pomace, rather than performing a comprehensive physicochemical characterization of the raw material. Therefore, parameters such as moisture content, particle size distribution, and SEM observations were not included in this manuscript.

Comments 3: Conventional drying at 34°C and freeze drying involves freezing at -80°C. These two methods are extremely different in terms of energy consumption and time, so please explain the scientific basis for choosing these two specific drying methods. 

Response 3: Thank you for your observation. We are aware of the significant differences between these methods; however, they were intentionally selected to compare two contrasting drying techniques commonly applied to fruit by-products. The scientific rationale and discussion of their impact on extraction efficiency are provided in the Discussion section.

Comments 4: The basis for the selection of the pressure range (300-500 bar) and the temperature range (30-50°C) is not adequately described, and it is recommended that literature be cited to support the rationale for these parameter ranges.

Response 4: We agree with this observation. In the revised manuscript, the rationale for selecting the pressure and temperature ranges has been elaborated, and relevant literature references supporting these choices have been added in Results section.

Comments 5: It is recommended that antioxidant activity determinations of extracts (e.g. DPPH, ABTS, etc.) be added .

Response 5: Thank you for this valuable suggestion. Antioxidant activity analysis was not within the scope of the present study, which focused primarily on extraction efficiency and phenolic content. However, we recognize its importance and plan to include antioxidant activity evaluations in future research.

Comments 6: The expression of numbers and units needs to be harmonized (e.g. spaces).

Response 6: Thank you for pointing this out. The entire manuscript has been carefully reviewed, and the expression of numbers and units has been harmonized according to journal guidelines.

Round 2

Reviewer 1 Report

Comments and Suggestions for Authors

Dear Editor,

I have carefully reviewed the revised version of the manuscript and confirm that the authors have satisfactorily addressed all minor corrections requested in my previous evaluation.

The authors have demonstrated diligence and attention to detail in implementing all requested corrections. The modifications are appropriate and improve the editorial quality of the manuscript without affecting the scientific content.

The revised manuscript now meets the standards of Applied Sciences and only requires minor formatting adjustments.

Terminology standardization

In line 57, the term “lyophilization” (with a z, as in American English) is used, while in line 95, “lyophilization” (with an s, as in British English) is used. It is important to maintain terminological consistency throughout the manuscript, using a single variant (either British or American) consistently throughout the text, in accordance with the journal's standard or preference.

Decimal separators

In Table 3, the results of the dependent variables must be corrected, as they contain commas instead of full stops. The English numerical format must be used.

The same applies to the text.

Line 288: “16,37” should be corrected to “16.37”

Line 294: “4,74” should be corrected to “4.74”

This change ensures consistency and coherence with the numerical style required by the journal.

References

Line 443: The reference number for “Kandemir et al. (2022)” should be corrected.

Line 446: The reference number for “Fomo et al. (2020)” should be corrected

Author Response

Thank you for your careful re-evaluation and valuable remarks. All suggested corrections have been implemented in the revised manuscript: the terminology has been unified, consistently using the form lyophilization throughout; numerical formatting in the text and tables has been corrected by replacing commas with full stops in accordance with the required style; and the numbering of the references to Kandemir et al. (2022) and Fomo et al. (2020) has been adjusted. The entire reference list has also been re-checked for accuracy and consistency. We believe that the revised version now fully meets the standards of Applied Sciences.

Reviewer 2 Report

Comments and Suggestions for Authors

In the second version of this paper the issues reported by me for the first version were addressed or received pertinent explanations:

  • The Abstract was completed with the most relevant results;
  • The necessity of a polar medium for extraction of total phenolic compounds (TPC) was discussed;
  • The Discussion section has been supplemented with relevant information that support the obtained results;
  • The TPC values have been recalculated and now fall within a plausible range for the considered type of raw material;
  • Some of the results have been expressed as average and standard deviation;
  • Some of the References were supplemented with the missing data.

However, there are still some small problems that have not been corrected:

  • The results from Table 6 should be expressed as average and standard deviation;
  • The References should be carefully checked because contain a lot of mistakes (for example: Ref. 1 was published in Current Opinion on Food Sciences not in Amsterdam; Ref. 2 was published in Journal of Supercritical CO2 Utilization not in Supercrit. Fluids; Ref. 3 was published in Plants not in Antioxidants: Ref. 4 was published in Engineering Proceedings not in Foods)
  • Line 66, 76, 77 - the Latin names must be written with Italic font.

Author Response

Thank you very much for your thorough re-evaluation and constructive remarks. We have carefully addressed all remaining issues in the revised manuscript: the results in Table 6 are now expressed as mean values accompanied by standard deviations; the entire reference list has been carefully re-checked and corrected, with the previously mentioned errors (Refs. 1–4) as well as other inconsistencies amended in line with the journal’s requirements; and the Latin names of species (lines 66, 76, and 77) have been formatted in italics. We believe that these corrections, together with the improvements already implemented in the previous revision, ensure that the manuscript now meets the standards of Applied Sciences.

Reviewer 3 Report

Comments and Suggestions for Authors

I have no further comments.

Author Response

We sincerely thank the Reviewer for their careful evaluation of our work and their positive conclusion that no further comments are required. We greatly appreciate the time and effort dedicated to reviewing our manuscript.

Reviewer 4 Report

Comments and Suggestions for Authors

1.Introduction Section: The last two paragraphs mention both the aim of the study and the objective of the study, which appears repetitive. Please revise for clarity and conciseness.

2.Abstract Section: The abstract requires revision to properly structure the background, methods, results, and conclusion.The first sentence does not clearly present the research background and should be rephrased for better context.

3.The first schematic figure is missing a caption. Please provide an appropriate label and description.

Author Response

Comments 1: Introduction Section: The last two paragraphs mention both the aim of the study and the objective of the study, which appears repetitive. Please revise for clarity and conciseness.

Response 1: The Introduction section has been revised to remove repetition and present the aim and objectives of the study more clearly and concisely

Comments 2: Abstract Section: The abstract requires revision to properly structure the background, methods, results, and conclusion.The first sentence does not clearly present the research background and should be rephrased for better context.

Response 2: The Abstract has been rewritten to follow a structured format (background, methods, results, and conclusions). The first sentence has been rephrased to better contextualise the research background.

Comments 3: The first schematic figure is missing a caption. Please provide an appropriate label and description.

Response 3: The first schematic figure has been supplemented with a caption and an appropriate description to ensure clarity and completeness.